# Comparison of Traditional and Ultrasound-Enhanced Electrospinning in Fabricating Nanofibrous Drug Delivery Systems

**DOI:** 10.3390/pharmaceutics11100495

**Published:** 2019-09-26

**Authors:** Enni Hakkarainen, Arle Kõrkjas, Ivo Laidmäe, Andres Lust, Kristian Semjonov, Karin Kogermann, Heikki J. Nieminen, Ari Salmi, Ossi Korhonen, Edward Haeggström, Jyrki Heinämäki

**Affiliations:** 1School of Pharmacy, University of Eastern Finland, 70210 Kuopio, Finland; hakkarainen.enni@gmail.com (E.H.); ossi.korhonen@uef.fi (O.K.); 2Institute of Pharmacy, Faculty of Medicine, University of Tartu, 50411 Tartu, Estonia; arle.korkjas@ut.ee (A.K.); ivo.laidmae@ut.ee (I.L.); andres.lust@ut.ee (A.L.); kristian.semjonov@gmail.com (K.S.); kkogermann@gmail.com (K.K.); 3Department of Immunology, Institute of Biomedicine and Translational Medicine, University of Tartu, 50411 Tartu, Estonia; 4Electronics Research Laboratory, Department of Physics, University of Helsinki, 00014 Helsinki, Finland; heikki.j.nieminen@aalto.fi (H.J.N.); ari.salmi@helsinki.fi (A.S.); edward.haeggstrom@helsinki.fi (E.H.); 5Medical Ultrasonics Laboratory, Department of Neuroscience and Biomedical Engineering, Aalto University, 02150 Espoo, Finland

**Keywords:** nanotechnology, nanofibers, traditional electrospinning, ultrasound-enhanced electrospinning, drug delivery system

## Abstract

We investigated nozzleless ultrasound-enhanced electrospinning (USES) as means to generate nanofibrous drug delivery systems (DDSs) for pharmaceutical and biomedical applications. Traditional electrospinning (TES) equipped with a conventional spinneret was used as a reference method. High-molecular polyethylene oxide (PEO) and chitosan were used as carrier polymers and theophylline anhydrate as a water-soluble model drug. The nanofibers were electrospun with the diluted mixture (7:3) of aqueous acetic acid (90% *v*/*v*) and formic acid solution (90% *v*/*v*) (with a total solid content of 3% *w*/*v*). The fiber diameter and morphology of the nanofibrous DDSs were modulated by varying ultrasonic parameters in the USES process (i.e., frequency, pulse repetition frequency and cycles per pulse). We found that the USES technology produced nanofibers with higher fiber diameter (402 ± 127 nm) than TES (77 ± 21 nm). An increase of a burst count in USES increased the fiber diameter (555 ± 265 nm) and the variation in fiber size. The slight-to-moderate changes in a solid state (crystallinity) were detected when compared the nanofibers generated by TES and USES. In conclusion, USES provides a promising alternative for aqueous-based fabrication of nanofibrous DDSs for pharmaceutical and biomedical applications.

## 1. Introduction

Electrospinning (ES) is a method for fabricating polymeric nanofibrous constructs, which have potential applications in pharmaceutical and biomedical fields. Nanofibers are typically tenth-to-hundred nanometers thick, they feature large outer surface, substantial surface- area-to-volume ratio, and high porosity (nanomats). This makes these fibers interesting for drug delivery and tissue engineering applications [1,2,3]. To date, nanofibers have found use, e.g., in formulation of poorly water-soluble drugs, fabrication of novel drug delivery systems (DDSs), supporting wound healing as wound dressings or artificial skin substitutes, and as scaffolds in tissue engineering [3,4,5].

ES has been applied as a manufacturing method in the clothing, electronics and optical industries, and during the past twenty years it has gained increasing interest in the pharmaceutical and biomedical industries. In traditional ES (TES), a polymer solution is first translated via a capillary tube to a spinneret and then spun towards a grounded collector plate or roll using a high-voltage electron field between the spinneret and collector [6,7]. The major limitations associated with the use of a simple single-fluid TES are blockage of a spinneret (nozzle) system, hazards related to the use of organic solvents (including residual solvent in the nanofibers), and long processing times. More recently, modified two-fluid and tri-fluid coaxial ES methods have been introduced to advance ES of even complicated nanostructures [8]. The clogging phenomena associated with ES can be eliminated by using such modified coaxial ES and concentric needle spinneret [8]. The morphology and diameter of TES nanofibers depend on the intrinsic properties of the solution, the type of polymer, conformation of the polymer chain, the viscosity, elasticity, electric conductivity, as well as the polarity and surface tension of the solvent [1,2,3,4]. In recent years, interest has been focused on developing nozzleless ES technologies to avoid the above-mentioned challenges related to TES.

Ultrasound-enhanced ES (USES) provides an orifice-free ES technique that employs ultrasound (US) to create nanofibers [9]. In this technique, high-intensity focused US bursts generate a liquid protrusion with a Taylor cone from the surface of an electrospinning solution (Figure 1). When the drug-polymer solution is charged with high negative voltage, a nanofiber jet is generated from the tip of the protrusion and this jet is led to an electrically grounded collector residing at a constant distance from the fountain [10]. The USES have some advantages over TES: the blockage of a spinneret system and the inclusion of hazardous organic solvents can be avoided with USES. In a USES setup, there is no nozzle that may clog and the evaporation of solvent is more efficient than in a TES setup. With USES, the evaporation of solvent is advanced by using a high-intensity focused US. A travelling US wave generates acoustic streaming inside the solution and induces thermal effect (heating) on the surface of the liquid, thus advancing the evaporation of the solvent. The generation of a liquid protrusion with a Taylor cone can be modified by changing US frequency, pulse repetition frequency and cycles per pulse [9,10].

In the present study, we compared the TES and USES techniques as means to fabricate drug-loaded polymeric nanofibers and we investigated the physicochemical and pharmaceutical properties of the produced nanofibers/nanofibrous DDSs. The influence of these two nanofabrication processes on the fiber formation, geometric fiber properties, surface morphology and physical solid-state properties of nanoconstructs were investigated. Special attention was paid to the formation and physical characterization of the drug-loaded nanofibers generated by the USES method.

## 2. Materials and Methods

### 2.1. Materials

Theophylline anhydrate (CAS No. 58-55-9; chemical purity ≥ 99%; Sigma-Aldrich Inc., Saint Louis, MO, U.S.A) was used as a water-soluble model drug. Polyethylene oxide, PEO (CAS No. 25322-68-3; Product No. 189456; average molecular weight 900,000 Da) and chitosan (CAS No. 9012-76-4; Product No. 448877; medium molecular weight grade) (Sigma-Aldrich Inc., Saint Louis, MO, U.S.A) were investigated as carrier polymers in both TES and USES nanofabrication. The diluted mixture (7:3) of aqueous acetic acid (CAS No. 64-19-7; chemical purity 99.9%) (90% *v*/*v*) and formic acid (CAS No. 64-18-6; chemical purity ≥ 98%; Ph. Eur., Strasbourg, France) solution (90% *v*/*v*) (with a total solid content of 3% *w*/*v*) was used as a solvent system for ES.

### 2.2. Fabrication of Nanoconstructs

The composition of the electrospun nanofibers is shown in Table 1. The nanofibers were generated in a TES (ESR-200Rseries, eS-robot^®^, NanoNC, Seoul, Korea) and in a custom-made in-house USES method. The USES method is described in detail in [8]. In brief, the USES setup features a vessel containing a spinning solution, a US generator and a transducer, a membrane system between the bottom of the vessel and the US transducer, a high-voltage electrode, and a grounded collector plate.

To modulate the fiber diameter, specific US parameters (frequency, pulse repetition frequency and cycles per pulse) were exploited in an ES process. Table 2 lists the process parameters applied to fabricate TES and USES nanofibers.

### 2.3. Characterization of Nanofibers

The nanofibrous samples were stored in a zip-lock plastic bag in ambient room temperature (22 ± 2 °C) prior to characterization. Scanning electron microscopy, SEM (Zeiss EVO MA15, Jena, Germany) and optical microscopy were applied to study fiber size distribution and morphology of nanofibers. The samples were coated with a platinum layer (6 nm) prior to imaging with SEM. Three SEM images of each sample were taken using three different magnifications (400×, 2000–2500× and 10,000×). ImageJ software Version 1.51K was used to measure the diameter of nanofibers. Statistical evaluation (*t*-test) was made using Microsoft Excel 2016 (Microsoft Corp., Albuquerque, NM, USA).

Physical solid-state and thermal properties were investigated by means of Fourier Transform Infrared (FTIR) spectroscopy (IRPrestige 21, Shimadzu corporation, Kyoto, Japan) with a single reflection attenuated total reflection (ATR) crystal (Specac Ltd., Orpington, UK), X-ray diffraction, XRD (Bruker D8 Advance diffractometer, Bruker AXS GmbH, Karlsruhe, Germany), and differential scanning calorimetry, DSC (DSC 4000, Perkin Elmer Ltd., Shelton, CT, USA). XRD and FTIR spectroscopy results were normalized and scaled. In all DSC experiments, the sample size was 3–6 mg. The samples were first cooled down and kept at 0 °C for three minutes, and then heated to 350 °C at a rate of 10 °C/min. The samples were then cooled to 0 °C (10 °C/min) and then heated to 350 °C (10 °C/min). The DSC thermogram for PEO was obtained by heating the sample from 30 °C to 170 °C with a heating rate of 10 °C/min. For solid-state characterization, the corresponding binary or ternary physical mixtures (PMs) were prepared manually with a mortar and pestle, and they were used as reference samples for the nanofibrous samples.

## 3. Results and Discussion

### 3.1. Topographical and Fiber Size Comparison of Nanoconstructs

Figure 2 illustrates the topography (surface morphology) of the polymeric nanofibrous constructs generated by TES (A, B) and USES (C, D). Figure 3 shows the comparison of the average diameter of individual nanofibers produced by TES and USES. With all fiber compositions tested, TES produced thinner and more uniform-by-size polymeric nanofibers in comparison with those generated by the nozzle-free USES technique. The diameter of nanofibers produced by TES was 77 ± 21 nm, and the diameter of the corresponding nanofibers generated by USES were 402 ± 127 nm (with a burst count of 400 cycles) and 555 ± 265 nm (with a burst count of 700 cycles). Statistically significant difference (*p*< 0.001) was shown between the fiber diameter of nanofibers obtained with TES and the nanofibers generated with USES. This difference in fiber size could be explained by the fact that the USES is a multivariate process involving an open vessel and more critical process parameters (including US parameters) to be controlled than in the TES. The sensitivity of the polymer solution to US and the variations in distance between the surface of the ES solution and the collector plate could be potential reasons for these differences. In aqueous polymer solution ES, the process and ambient parameters such as conductivity, applied voltage, relative humidity, and the distance between a nozzle tip and collector plate could affect the diameter of nanofibers (i.e., increasing the level of these parameters leads to generation of thinner fibers) [4]. However, in fabricating nanofibers for pharmaceutical and biomedical applications, having nanofibers as small as possible is not of intrinsic value in itself and is not necessarily an ultimate goal. For example, in wound healing and many tissue engineering applications, a fiber size close to the micron-scale is considered beneficial in terms of cell adhesion and proliferation [11].

With USES, the fiber diameter can be modulated by changing the burst count (cycles per US pulse = duty factor). As seen in Figure 3, changing a burst count from 400 to 700 cycles generated nanofibers with the average diameter of 402 nm and 555 nm, respectively (the other critical US parameters, i.e., frequency and pulse repetition frequency, were kept constant). The statistical analysis showed that the diameters of USES nanofibers generated with the two burst count cycles were different (*p* < 0.001). In the TES, solutions with high conductivity and high surface tension require high voltages and the change in applied voltage (electric field) has only a minor effect on fiber diameter of the nanofibers [4]. Therefore, the process flexibility of USES (i.e., the dynamic modulation of fiber size) is an advantage over TES. The use of higher voltages in the TES increases also the risk of “bead” formation (= defects) in the nanofibrous mats due to the instability of a Taylor cone [4]. As seen in Figure 2, the USES nanofibrous constructs can be generated without signs of “beads” in the final nanofibrous mat. This is advantageous since the formation of “beads” is considered to be a sign of improper ES process.

### 3.2. Characterization of Nanoconstructs

#### 3.2.1. X-Ray Diffraction

The carrier polymer PEO has two typical diffraction peaks (2*θ*) at 19° and 23° [12]. As seen in Figure 4, both characteristic diffraction peaks of PEO are visible in the XRD patterns of the PEO powder and in the PM, thus revealing the semi-crystalline nature of PEO. According to the literature, crystalline theophylline gives several characteristic reflections at diffraction angles (2*θ*) at 7.2°, 12.6°, 14.3°, 24.1°, 25.6°, 26.4° and 29.4° [13]. The major reflection (2*θ*) is located at 12.6°. Crystalline chitosan has also two characteristic reflections (2*θ*) at approximately 10° and 20° [14].

The XRD patterns for the nanofibers produced by TES and USES appear nearly identical, and the characteristic diffraction peaks for the three pure materials can be distinguished (Figure 4). Both carrier polymers PEO and chitosan preserved their semi-crystallinity and crystallinity, respectively. We found differences between the diffraction patterns of nanofibrous samples fabricated by these two methods. As seen in Figure 4, nanofibers produced with TES displayed a slightly shifted diffraction peak at 6.6° 2*θ* which suggests a solid-state change in theophylline. We also found that this reflection and the characteristic peak of theophylline (at 7.2° 2*θ*) are absent in the XRD patterns of nanofibers generated by USES. Moreover, the other characteristic reflection (2*θ*) of theophylline at 12.8° 2*θ* is not distinguished in the XRD patterns of nanofibers (a new reflection shows less intensity and it is shifted to 13.5° 2*θ*). Therefore, it is evident that solid-state (crystallinity) changes in theophylline have taken place during the TES and USES nanofabrication. It is possible that in addition to an amorphous form, theophylline monohydrate or metastable theophylline or even the mixture of different forms may appear during or immediately after ES. Furthermore, all diffraction peaks in the XRD pattern of the nanofibers generated by USES appear weaker and less sharp than the corresponding reflections of the XRD pattern for the nanofibers produced by TES (Figure 4). This can be seen with the diffraction reflection (2*θ*) 19.2° which originates from PEO. With nanofibers generated by USES, the diffraction reflections (2*θ*) characteristic to semi-crystalline PEO at 19.2° and 23.3° are seen as slightly weaker than those in the XRD pattern of PM (Figure 4). These differences in the XRD patterns reveal that there is a difference in the crystallinity of nanofibers fabricated with the different methods. Application of high-intensity focused US in the USES process affects the solid-state properties of the nanofibers resulting in more amorphous (less ordered) nanostructures than those obtained with TES.

#### 3.2.2. Differential Scanning Calorimetry

The thermal behavior (DSC thermograms) of the pure materials, PM, and nanofibers produced by TES and USES are shown in 5 and 6. As seen in Figure 5, the melting endotherms for PEO and theophylline are at 70 °C and at 270 °C, respectively. The characteristic melting endotherm of PEO is seen in the DSC thermograms of PM (Figure 5) and in the thermograms of nanofibers generated by TES and USES (Figure 6). Chitosan as a pure material exhibited a broad endothermic event at 40–120 °C (due to water evaporation) and an exothermic event at 300 °C (chemical degradation) [15]. The melting of theophylline cannot be seen in the DSC thermograms of PMs and nanofibers due to the melting of polymer at lower temperatures and subsequent dissolution of theophylline in the molten polymer (PEO, chitosan). Hence, the DSC results cannot reveal whether theophylline exists in a crystalline form or an amorphous form in the nanofibers generated by TES or USES. However, the XRD patterns shown previously confirmed the solid-state of the drug (crystalline form rather than amorphous form) in the nanofibers generated by USES.

The DSC profiles of drug-loaded nanofibers generated by TES or USES were nearly identical suggesting that applying focused high-intensity US in the USES process does not significantly affect the solid-state properties of the nanofibers (Figure 6). As shown in Figure 6, the lower peak height of a characteristic melting endotherm for PEO at 70 °C indicates lower enthalpy of transition (∆H) with the nanofibers produced by TES than that of the fibers produced by USES. In the DSC thermograms of nanofibers generated by both TES and USES, a small exothermic event at 270 °C is seen, which is probably caused by the chemical decomposition of chitosan.

In the cooling phase, only the crystallization of PEO is observed in the DSC thermogram of the F-III nanofibers generated by USES (Figure 6). During re-heating of the sample (F-III), the thermal event (melting endotherm) of PEO is seen. Chitosan decomposed during the first heating but the fate of theophylline is not clear. No visible thermal events nor signals were detected in the DSC thermogram of pure theophylline after the first heating and no decomposition during heating nor any solidification during the cooling (Figure 5). The relatively small amount of theophylline (13%) in the PM and nanofibers may explain why the melting of theophylline is not recognized in the DSC thermograms of PM or nanofibers, but it cannot account for the lack of thermic events during the cooling phase or in the second heating of pure theophylline. Crystallization of theophylline during the DSC cooling phase has been reported previously [13,16].

#### 3.2.3. Fourier Transform Infrared (FTIR) Spectroscopy

Figure 7 shows FTIR spectra for the pure materials, a physical mixture (PM) and the nanofibers produced by TES and USES. No significant changes in chemical structure of the materials were found in the nanofibers generated with TES or the USES nanofabrication process. The characteristic peaks for theophylline were identified in both TES (F-I) and USES (F-II, III) nanofibrous constructs. The FTIR spectra were close to identical for both types of nanofibers. A characteristic peak of PEO at 2875 cm^−1^ presents the stretching of C–H [17]. It can be observed in all FTIR spectra except in the spectrum for chitosan and theophylline (Figure 7). The characteristic absorption peaks of the theophylline spectrum at 1665–1550 cm^−1^ are derived from C–C and C–N bonds [18]. We found small and characteristic absorption peaks for chitosan (as a pure material) and a peak with a higher intensity for the PM in the same spectral region (Figure 7). This specific peak (1658 cm^−1^) is partially visible in the FTIR spectra of the nanofibers produced by TES (F-I) and USES (F-II, III). The intensity of the characteristic peak of theophylline at 1658 cm^–1^ for nanofibrous samples (F-II, F-III), however, is smaller than that observed in the FTIR spectra of PM or pure theophylline powder.

## 4. Conclusions

We compared traditional electrospinning (TES) and nozzleless ultrasound-enhanced electrospinning (USES) as methods to fabricate nanofibrous polymeric drug delivery systems (DDSs). The physicochemical and pharmaceutical properties of the nanofibrous DDSs were studied. Both methods can be applied for aqueous-based fabrication of non-woven DDSs using PEO and chitosan as carrier polymers. With USES, the evaporation of solvent is advanced by using a high-intensity focused US enabling acoustic streaming and thermal effect inside the solution. Therefore, USES is associated with more pronounced process-induced solid-state changes of the materials compared to those induced by TES. Nanofibers generated by USES are amorphous, whereas the nanofibers produced by TES are less prone to being amorphous. The controlled phase transformation of higher-energy amorphous form is especially poorly applicable for water-soluble drugs. Further research is needed to discover all potential strengths and limitations of USES in fabricating nanofibrous DDSs.

## Figures and Tables

**Figure 1 pharmaceutics-11-00495-f001:**
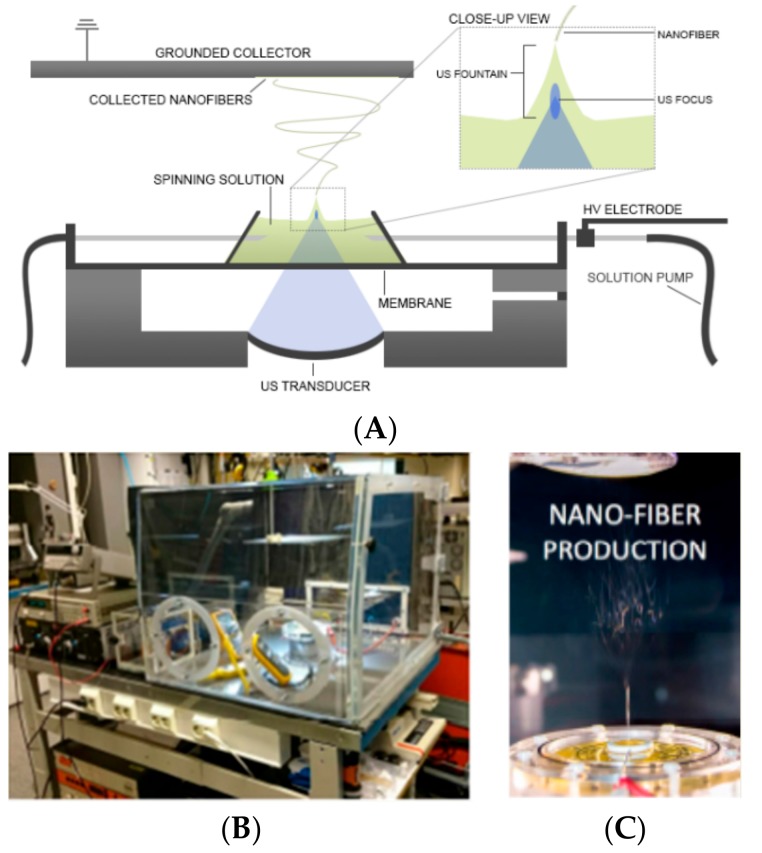
(**A**) Schematic diagram of the ultrasound-enhanced electrospinning (USES) setup, (**B**) photograph of the USES system and process environment (including a humidity cabinet), and (**C**) close-up photograph on the formation of nanofibers in a USES process.

**Figure 2 pharmaceutics-11-00495-f002:**
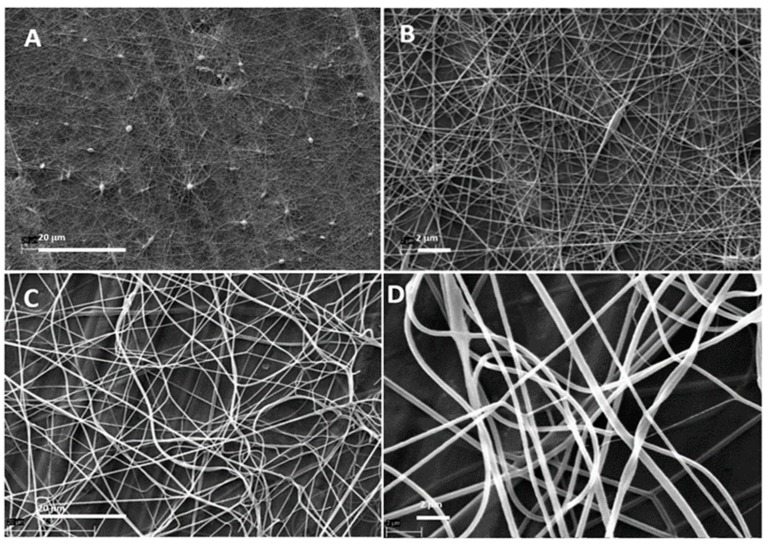
Scanning electron microscopy (SEM) images of traditional electrospun (TES) and ultrasound-enhanced electrospun (USES) nanofibers. (**A**,**B**) TES nanofibers (magnification 2500× and 10,000×); (**C**,**D**) USES nanofibers (2500× and 10,000×).

**Figure 3 pharmaceutics-11-00495-f003:**
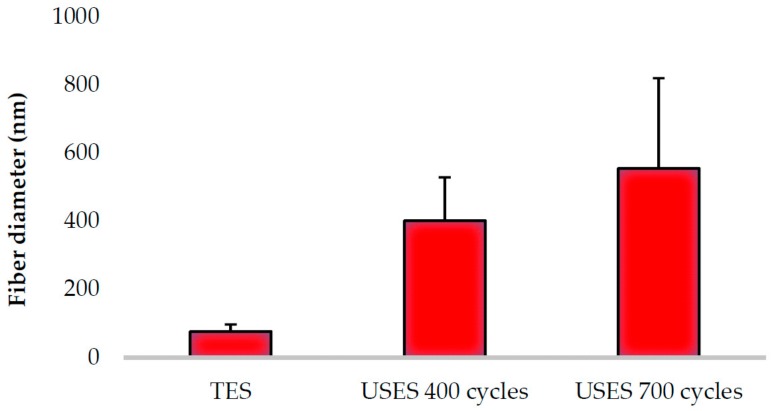
Average diameter (mean ± SD) of traditional electrospun (TES) and ultrasound-enhanced electrospun (USES) nanofibers. The fiber size analysis is based on three SEM images, and the total number of analyzed individual nanofibers was *n* = 100 (with USES 700 cycles *n* = 53). The number of cycles in the USES process (US signal) refers to pulse duration.

**Figure 4 pharmaceutics-11-00495-f004:**
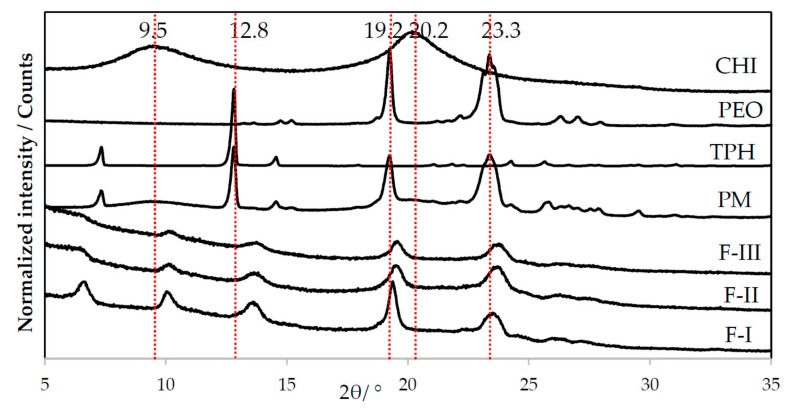
Normalized X-ray diffraction (XRD) patterns of pure materials, physical mixture (PM) of drug and carrier materials, and nanofibers generated with traditional electrospinning, TES (formulation F-I), and ultrasound-enhanced electrospinning, USES (F-II, F-III). Key: CHI = Chitosan, PEO = Polyethylene oxide, TPH = Theophylline anhydrate form II.

**Figure 5 pharmaceutics-11-00495-f005:**
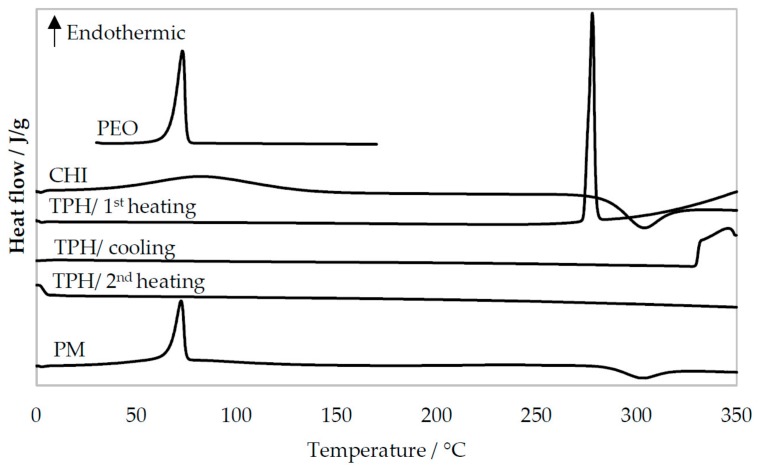
Differential scanning calorimetry (DSC) thermograms of pure materials and physical mixture (PM) of drug and carrier materials. For PEO, CHI, and PM, only the first heating is presented. Key: CHI = Chitosan, PEO = Polyethylene oxide, TPH = Theophylline.

**Figure 6 pharmaceutics-11-00495-f006:**
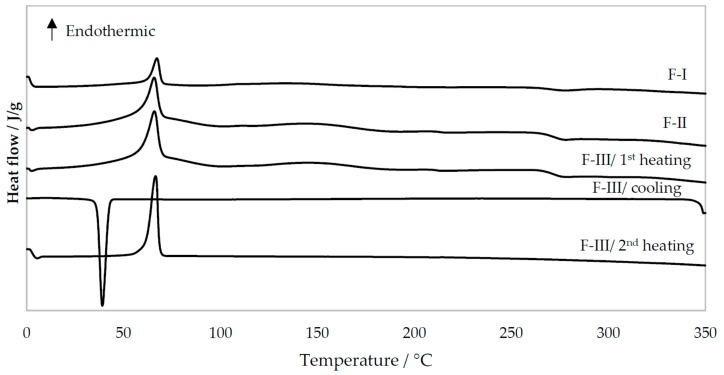
Differential scanning calorimetry (DSC) thermograms of nanofibers generated with traditional electrospinning, TES (formulation F-I) and ultrasound-enhanced electrospinning, USES (F-II, F-III). For F-I and F-II only the first heating is presented. Key: CHI = Chitosan, PEO = Polyethylene oxide, TPH = Theophylline.

**Figure 7 pharmaceutics-11-00495-f007:**
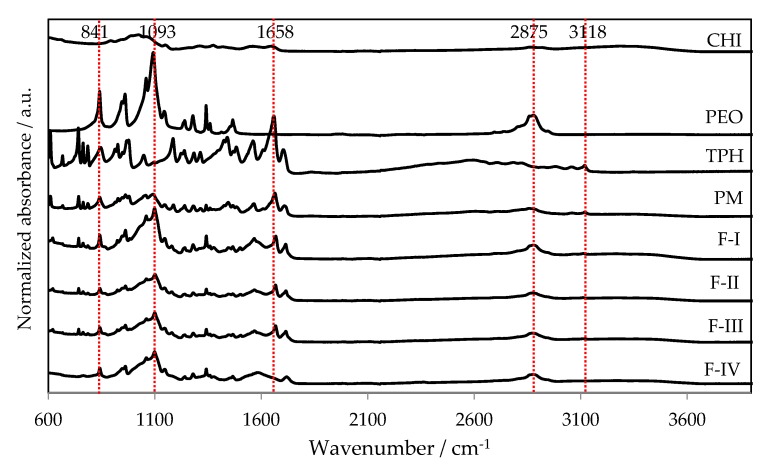
Normalized FTIR spectra for pure materials, the physical mixture (PM) of drug and carrier materials, and nanofibers generated with traditional electrospinning, TES (formulation F-I) and ultrasound-enhanced electrospinning, USES (F-II, F-III and F-IV). Key: CHI = Chitosan, PEO = Polyethylene oxide, TPH = Theophylline.

**Table 1 pharmaceutics-11-00495-t001:** Composition (% *w*/*w*) of nanofibers generated by traditional electrospinning (TES) (I) and ultrasound-enhanced electrospinning (USES) (II–IV.)

Formulation/Ingredient	I	II	III	IV
Chitosan	43.5	43.5	34.8	40
Polyethylene oxide (PEO)	43.5	43.5	52.2	60
Theophylline	13.0	13.0	13.0	0

**Table 2 pharmaceutics-11-00495-t002:** Process parameters applied in the traditional electrospinning (TES) (I) and ultrasound-enhanced electrospinning (USES) (II–IV) of nanofibers.

Formulation/Parameter	I	II	III	IV
Voltage (kV)	11.5–14.0	16.0	16.0	14.0–16.0
Voltage of collector (kV)	NA	−5.0	−5.0	−5.0
Distance (cm)	15.0	17.0	17.0	17.0
Pumping rate (mL/h)	0.3	0.8	0.6	0.6
Amplitude (mV)	NA	250	240	200–240
Frequency (MHz)	NA	2.06	2.06	2.06
Burst count (Cycles)	NA	1000	1000	1000
Burst rate (Hz)	NA	70	70	70
Humidity (RH%)	18	19	24	30

NA = not applicable.

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
