# Peer review of "Comparison of Traditional and Ultrasound-Enhanced Electrospinning in Fabricating Nanofibrous Drug Delivery Systems"

_pharmaceutics, 2019, doi:10.3390/pharmaceutics11100495_

Round 1

Reviewer 1 Report

This manuscript deals with a new technique of ultrasonic nozzles usable for electrostatic spinning, which was patented by the authors in previous years. The mechanical capillary needle nozzle (known from TES) has been replaced by an ultrasonic generator (USES), which on the surface of the solution helps the formation of Taylor cones. The authors further show the advantages of this arrangement by analysis of composite Chitosan/PEO/Theophylline nanofibrous layers which are researched for medical applications. The manuscript is well organized, readable and error free. At the same time, it brings new knowledge that enriches the whole field of research. I agree with the publication of this manuscript as soon as the following statements are clarified.

Line 48: How the new setup solves TES disadvantages? Especially the process time and the use of organic solvents? Using USES, the production rate did not increase according to the results, the solvent evaporates from the free liquid surface of the USES nozzle and thus the process is more hazardous than TES (see line 60 as well) also the spinning solution concentration increases with the deposition time. At line 61: Explain or refer to why solvent evaporation is more effective in case of USES? The statements in lines 155 and 156 (even so 269 and 270) are not supported by experimental results. In Figure 3, fiber diameters are not statistically different for different cycle numbers. Please give the appropriate justification. Please, use “fiber diameter” instead of “fiber thickness”.

Author Response

Response to Reviewer 1 Comments

Point 1: Line 48: How the new setup solves TES disadvantages? Especially the process time and the use of organic solvents? Using USES, the production rate did not increase according to the results, the solvent evaporates from the free liquid surface of the USES nozzle and thus the process is more hazardous than TES (see line 60 as well) also the spinning solution concentration increases with the deposition time.  

Response 1: The primary advantage of USES over TES is that there is no risk of the blockage of a spinneret system, since USES provides a nozzle-free ES approach based on acoustic radiation pressure. We also showed that the ES of aqueous polymeric solutions is doable with USES (due to the enhanced evaporation of solvent under a high-intensity focused US). The use of aqueous or ethanol solutions in TES is challenging due to the limited evaporation capacity of such solutions. Therefore, we see that USES could provide an alternative ES technology for replacing hazardous and highly volatile (or flammable) organic solvents in ES. Due to the unique working principle, USES could also enable gradients in fiber constructs, which is not possible with TES. We agree with the Reviewer´s remark that there are not any significant differences between the production rate of USES and TES shown in the present work. However, we do believe that by further developing a USES technology, we could improve the production rate by optimizing the process conditions and scaling up. For example, the vessel containing spinning solution can be equipped with multiple parallel US generators and transducers for the generation of nanofibers. The increase of the spinning solution concentration on the course of a USES process is also a relevant issue (especially with highly volatile organic solvents), and it would be indeed the topic area of interest for the further development of USES. At the moment, we can cover the vessel applied in a USES setup to limit the “side-evaporation” and concentration increase of a spinning solution. We have revised the manuscript text to clarify for the readers the potential advantages of USES over TES.      

Point 2: At line 61: Explain or refer to why solvent evaporation is more effective in case of USES?

Response 2: With USES, the evaporation of solvent is contributed by using a high-intensity focused ultrasound (US). A travelling US wave generates acoustic streaming inside the solution and induces thermal effect (heating) on the surface of liquid, thus advancing the evaporation of solvent. In USES, the generation of a liquid protrusion with a Taylor cone can be modified (enhanced) by changing US characteristics such as frequency, pulse repetition frequency and cycles per pulse. To explain this solvent evaporation aspect for the readers, the manuscript has been revised accordingly.

Point 3: The statements in lines 155 and 156 (even so 269 and 270) are not supported by experimental results.

Response 3: We agree with your remark and have omitted the statements in lines 155 and 156 (as well as 269 and 270) as requested.

Point 4: In Figure 3, fiber diameters are not statistically different for different cycle numbers. Please give the appropriate justification. Please, use “fiber diameter” instead of “fiber thickness”.

Response 4: Statistical significant difference (p<0.001) was shown between the fiber diameter of nanofibers obtained with TES and the nanofibers generated with USES. Moreover, the diameters of USES nanofibers fabricated by different burst count cycles were also statistically different (p<0.001). The manuscript text has been revised accordingly and the wording “fiber thickness” has been replaced by “fiber diameter”. Statistical evaluation (t-test) was made using Microsoft Excel 2016 (Microsoft Corp., USA).

Reviewer 2 Report

The paper by Hakkarainen et al. deals with the Comparison of Traditional and Ultrasound-Enhanced Electrospinning in Fabricating Nanofibrous Drug Delivery Systems. This work is a good contribution to the field and could be published in Pharmaceutics after major revision as mentioned below:

CAS number and purities of all chemicals should be added in the material section Scale bar is not visible in Figure 2. Please add it manually Authors claim in the end of their introduction that they determine surface topography (AFM?) and mechanical properties (tensile measurement?) of the fibers; can the authors add these characterizations to the paper please Can the authors determine the difference in drug releasing (Theophylline anhydrate?) between TES and USES fibers?

Author Response

Response to Reviewer 2 Comments

Point 1: CAS number and purities of all chemicals should be added in the material section.

Response 1: The CAS number and purities of chemicals have been added in the manuscript (cf. the revised manuscript).  

Point 2: Scale bar is not visible in Figure 2. Please add it manually.

Response 2: We have revised Figure 2 by improving the visibility of scale bars (cf. the revised manuscript). 

Point 3: Authors claim in the end of their introduction that they determine surface topography (AFM?) and mechanical properties (tensile measurement?) of the fibers; can the authors add these characterizations to the paper please.

Response 3: We do apologize for this mistake. The phrase “surface topography” is somewhat misleading in this context, since the physical appearance and surface texture of the nanofibrous mats were characterized only by means of SEM (not by AFM or 3D laser profilometer). Therefore, we have revised the end of introduction section accordingly. Regarding with mechanical properties, we made a number of attempts to analyse mechanical strength of the nanofiber mats by means of texture analysis, but the results were not very convincing. Therefore, the results of the texture analysis were decided to leave out from the present paper. Unfortunately, in the manuscript text the phrase “mechanical properties” was erroneously remained. The manuscript has been now revised by re-phrasing the aims of the study in the end of introduction section (i.e., the phrase “mechanical properties” was omitted).      

Point 4: Can the authors determine the difference in drug releasing (Theophylline anhydrate?) between TES and USES fibers?

Response 4: This would be indeed a relevant response to be investigated with the present nanofibrous systems in the future. In this study, however, we focused at comparison of the traditional (TES) and a novel nozzle-free ultrasound-enhanced electrospinning (USES) processes in fabricating polymeric nanofibers. A major emphasis was laid on the fiber formation, physical appearance, geometric fiber properties, surface morphology and physical solid-state properties of nanoconstructs (i.e., potential process-induced transformations). Further research is needed to discover all potential strengths and limitations of USES in fabricating nanofibrous drug delivery systems (including drug release properties). This will be one of the key topic areas in our future studies on USES.

Reviewer 3 Report

The manuscript reports the comparisons between the traditional electrospinning (TES) and nozzleless ultrasound-enhanced electrospinning (USES)  in creating medicated nanofibers. Although USES has been reported previously, the present manuscript still has its merits to support its acceptance for publication in PHARMACEUTICS. The following suggestions please to be considered for improving the manuscript’s quality.

There should be some quantitative results in the ABSTRCAT. In the INTRODUCTION section, the first sentence should be “Electrospinning (ES) is a method for fabricating polymeric nanofibrous constructs, which have potential applications in pharmaceutical and biomedical fields”. Line 41 [3,5] should be [3-5]. Line 47-49, the sentence should be revised because modified coaxia electrospinning (using concentric needle spinneret, e.g. Polymers, 2019, 11, 1287) can be easily eliminate the clogging phenomena. A full background and some comments on modified coaxial/tri-axial electrospinning should be added here for the readers. It is better to conduct the in vitro drug dissolution tests to detect the drug release profiles from the prepared nanofibers. The scale bars in Figure 2 are too small, please enlarge. The right arrangement should be the order of XRD-DSC and then FTIR, thus is because the former methods are utilized to disclose the physical state of components, the latter method is exploited to disclose the possible secondary interactions for forming amorphous nano composites.   

Author Response

Response to Reviewer 3 Comments

Point 1: There should be some quantitative results in the ABSTRCAT. 

Response 1: We have revised the ABSTRACT by adding the quantitative results on the fiber diameter of both TES and USES electrospun nanofibers (cf. the revised manuscript). 

Point 2: In the INTRODUCTION section, the first sentence should be “Electrospinning (ES) is a method for fabricating polymeric nanofibrous constructs, which have potential applications in pharmaceutical and biomedical fields”.

Response 2: The present phrase has been revised as suggested (cf. the revised manuscript).

Point 3: Line 41 [3,5] should be [3-5].

Response 3: This point has been revised accordingly (cf. the revised manuscript). 

Point 4: Line 47-49, the sentence should be revised because modified coaxial electrospinning (using concentric needle spinneret, e.g. Polymers, 2019, 11, 1287) can be easily eliminate the clogging phenomena. A full background and some comments on modified coaxial/tri-axial electrospinning should be added here for the readers.

Response 4: We do agree with your remark on the use of concentric needle spinneret in modified coaxial electrospinning to eliminate clogging phenomena. The introduction section has been revised by adding a short description on the modified coaxial / tri-axial electrospinning and a corresponding literature reference.

Point 5: It is better to conduct the in vitro drug dissolution tests to detect the drug release profiles from the prepared nanofibers.

Response 5: This would be indeed a relevant response to be investigated with the present nanofibrous systems in the future. In this study, however, we focused at comparison of the traditional (TES) and a novel nozzle-free ultrasound-enhanced electrospinning (USES) processes in fabricating polymeric nanofibers. A major emphasis was laid on the fiber formation, physical appearance, geometric fiber properties and orientation, and physical solid-state properties of nanoconstructs (i.e., potential process-induced transformations). Further research is needed to discover all potential strengths and limitations of USES in fabricating nanofibrous drug delivery systems (including drug release properties). This will be one of the key topic areas in our future studies on USES.

Point 6: The scale bars in Figure 2 are too small, please enlarge.

Response 6: We have revised Figure 2 by enlarging the display of scale bars (cf. the revised manuscript). 

Point 7: The right arrangement should be the order of XRD-DSC and then FTIR, thus is because the former methods are utilized to disclose the physical state of components, the latter method is exploited to disclose the possible secondary interactions for forming amorphous nano composites.   

Response 7: We do agree with your remark on the relevant order of the XRD, DSC and FTIR spectroscopy. The manuscript has been revised accordingly.

Round 2

Reviewer 3 Report

The manuscript's quality has been improved substantially.

I recommend its acceptance for publication in its present form.